# Study of Plasma-Based Vortex Generator in Supersonic Turbulent Boundary Layer

Pavel Polivanov *, Oleg Vishnyakov and Andrey Sidorenko

Khristianovich Institute of Theoretical and Applied Mechanics SB RAS, Institutskaya, 4/1, Novosibirsk 630090, Russia; sindr@itam.nsc.ru (A.S.)
* Correspondence: polivanov@itam.nsc.ru

**Abstract:** The problem of flow control under conditions of a turbulent boundary layer at transonic and supersonic free-stream velocities is considered. Such flows are integral components of the flight process and exert significant effects on the flow around both the aerodynamic object as a whole and its individual elements. The present paper describes investigations of a combined control device ("plasma wedge"), which is a wedge mounted along the flow with the energy supply at one side of the wedge owing to a spark discharge. The strategy of flow control by this device is based on increasing the momentum in the boundary layer, which enhances its resistance to the adverse pressure gradient and, as a consequence, its resistance to flow separation further downstream. The study includes experimental and computational aspects. The examined flow evolves on a rectangular flat plate with a sharp leading edge at the free-stream Mach number M = 1.45 and unit Reynolds numbers $Re_1 = 11.5 \cdot 10^6$ 1/m. The experiments are performed to study the velocity fields and the pressure distribution in the wake behind the actuator. The results show that a streamwise vortex is formed in the wake behind the actuator when the discharge is initiated. Reasonable agreement of the experimental data with numerical simulations allows one to conclude that the Reynolds-averaged Navier–Stokes equations are suitable tools for solving the problem considered.

**Keywords:** flow control; vortex generators; plasma actuators; supersonic flow; turbulent boundary layer





## 1. Introduction

The problem of flow separation control in a turbulent boundary layer is important for numerous applications of gas dynamics [1–3]. Such flows often arise near external and internal surfaces of various flying aerodynamic objects. A popular method of preventing boundary layer (BL) separation is intensification of momentum exchange between the near-wall and external flows owing to vortex generation. A device used for this purpose is usually the classical mechanical vortex generator whose height is of the order of the BL thickness δ. The first researcher who used vortex generators was Taylor [4]. Those vortex generators were extensively studied and approved both in experiments and in applications [5–9]. The simplicity of vortex generators has led to their popularity, which leads to widespread use not only in the aviation industry, but also in other areas, for example, in the automotive industry [10]. The review papers [11,12] describe the effect of vortex generators on separated flows at supersonic Mach numbers. Their undoubted advantages are good operational performance, reliability, and structural simplicity. However, their obvious drawbacks (generation of significant additional drag and inability to change the performance intensity) stimulated researchers to seek for alternative methods of flow separation control. Micro-vortex generators with heights smaller than δ generating streamwise Taylor–Gortler vortices in the boundary layer were proposed in the 1970s [13]. These vortex generators produce a smaller drag force as compared to the classical generators [14]. However, the vortices are also weaker; for this reason, vortex generators of this class have certain constraints on their positioning with respect to the separation line [12,15,16]. Similar

to the classical vortex generators, micro-vortex generators cannot be deactivated. The advantages and disadvantages of vortex generators are basically common to most passive flow control methods. For example, in paper [17], a method is considered for suppressing the separation flow by bypassing air from the windward side of the wing to the leeward side. The papers [18,19] consider a way to control a laminar separation bubble through the use of a local flexible membrane. As in the case of vortex generators, the disadvantage of such methods is the inability to deactivate these devices.

Another alternative action on the boundary layer is the use of jet vortex generators and other active methods based on energy supply to the flow. The idea of using jet vortex generators relies upon the generation of vorticity in the boundary layer impinging on a normal or oblique gas jet. This method is fairly efficient but has also some constraints caused by the complexity of engineering devices. A review of results on using jets for flow separation control can be found, e.g., in [20]. Apart from usual jets, the so-called synthetic jets or actuators with a zero-mass flux have been recently considered [21]. A device of this type is a cavity with a vibrating wall connected to the surface by a narrow channel. The main advantage of such devices is the simplicity of their structure because a supply pipeline is not needed.

A device implementing a modification of this method is a pulsed plasma synthetic jet actuator (PPSJA), where the pressure oscillations in the chamber are initiated and sustained by periodic electric discharges. This method was originally developed to excite perturbations in the laminar boundary layer, and it was devised by studying the stability of the layer [22]. It was demonstrated in [23,24] that the PPSJA-based control methods can affect the size of separation as well as the position and oscillations of the shock wave, which have a significant effect on separation in high-velocity flows.

There are recent investigations on using plasma methods of flow separation control, namely, methods directly using surface discharges for vorticity generation in the boundary layer. A review of these studies for the laminar boundary layer can be found in [25–29]. The operation principle of such devices is usually based on flow acceleration by the ion wind excited in the plasmas of various electric discharges [30,31]. However, the ion wind intensity is rather low, and it seems problematic to increase it because electrostatic forces both accelerate and decelerate the gas flow during the period of voltage oscillations [32]. In the turbulent flow, velocity oscillations are fairly significant, and the velocity increment in the near-wall layer due to the ion wing is rather small on the background of these oscillations. For this reason, methods based on different properties of the electric discharge are used for turbulent flows, e.g., the nanosecond dielectric barrier discharge (ns-DBD) and the localized arc filament plasma actuator (LAFPA).

The use of the ns-DBD for flow separation control in wide ranges of Mach and Reynolds numbers was considered in [33–35]. The main mechanisms of this actuator operation include interactions of shock waves arising due to instantaneous heating of the gas in the discharge region and a local increase in the mean temperature of the flow. With an appropriate choice of parameters, the first mechanism can ensure the desired effect and slightly reduce the separation region, as was shown in [36]. However, mean heating of the flow exerts an adverse effect by displacing the flow and, correspondingly, decreasing the angular momentum in the boundary layer, resulting in separation region extension.

In the case of LAFPA, the action is also performed due to intense local heating of the gas and is accompanied by the formation of shock waves induced by a large temperature gradient in the discharge region. The effect of these actuators can be explained in a simplified manner as the formation of virtual roughness on the surface (owing to local displacement of the gas because of its heating); the flow around these roughness elements becomes disturbed, resulting in vortex formation among other effects.

Plasma actuators can insert control disturbances in a wide range of frequencies and also participate in systems with feedback, which may be useful in flows containing unsteady regions [37–43]. The main advantage of plasma actuators is the fact that they do not affect the flow if they are not activated [44,45]; thus, these actuators can be located in various

places. Plasma devices can also be used to generate jets [46]. Moreover, only those devices are activated that are needed in a particular flight mode [47]. Plasma actuators are being actively studied for the problem of combustion stabilization [48,49]. The use of plasma actuators makes it possible to directly accelerate the flow in the desired region, for example, using MHD force [50]. The common drawback of these techniques is their low energy efficiency because the major fraction of energy is spent on gas heating. Energy expenses needed for effective flow control may be rather significant in the case of the turbulent boundary layer and transonic or supersonic flow.

The method considered in the present study refers to active control methods and combines mechanical and energetic actions. The control device (actuator) is a streamwise mounted wedge (fin) with the energy supply at one side due to a spark discharge. Its photograph and operation principle are shown in Figures 1 and 2. The actuator is a thin ceramic fin mounted streamlined to the flow. When the electric discharge is excited, a pressure difference is formed at the trailing edge of the fin and, as a consequence, a transverse velocity gradient arises. This leads to the generation of a vortex downstream. The actuator is entirely submerged into the boundary layer and splits the flow into two parts by a symmetric fin. One part of the flow (in our case, the left part) reaches the trailing edge of the fin virtually unaffected. At the same time, the other part of the flow (the right one) flows through the plasma region of the discharge where the gas temperature and density change. It was shown in [51] that the flow around a plasma discharge is similar to the flow around a semi-infinite body, resulting in an increase in the displacement thickness and a change in the velocity profile. As a result, near the trailing edge of the fin, the right and left streamlines have different velocity and pressure. This leads to the flow from the side of the discharge to the opposite side, resulting in vortex formation.

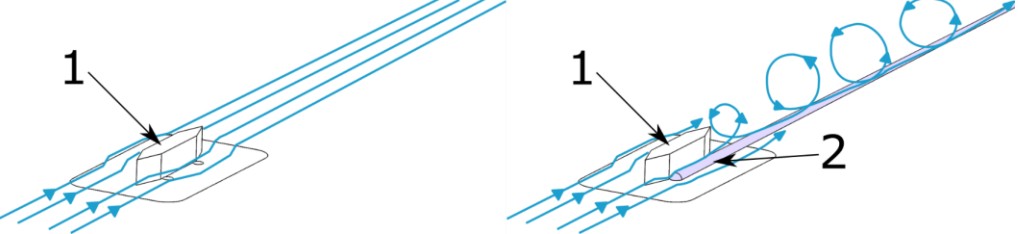

**Figure 1.** Scheme of actuator operation principle (1—wedge; 2—semi-infinite body).

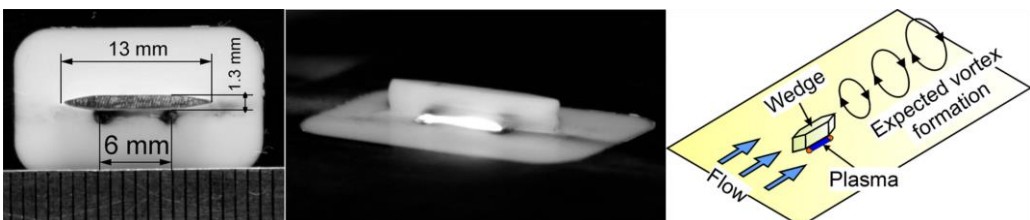

**Figure 2.** Photograph of the actuator and its operation principle.

If the actuator is in the passive state, it is expected to exert a minor effect on the flow because it has a small height (about 0.4–0.8δ) and a small cross-flow size. Thus, the actuator in the passive state generates a smaller drag force than the classical vortex generators or micro-generators. Certainly, from the viewpoint of additional drag generation, this actuator is less beneficial than other active methods, such as the surface discharge or jet vortex generator, which practically do not affect the flow being switched off. On the other hand, this device in the active state is less effective than the classical vortex generators and micro-generators; however, it is expected to be less energy expensive than surface energy supply and jet vortex generators.

Such a device was tested for the first time in [52,53], where transonic buffeting control was studied. It was found that the plasma wedge actuator can affect shock wave oscillations

and also the pressure distribution in the wake, though no data were obtained to verify the mechanism of its action on the flow. Numerical simulations performed in [54] were used as a basis for the development of the plasma wedge.

However, the effect of flow control found in papers [52,53] was small. Moreover, the calculations performed in paper [54] do not clarify the physics of the appearance of a streamwise vortex. To improve the effect of the generation of a streamwise vortex by a plasma wedge, it is necessary to carry out additional studies that will make it possible to clarify the process of the onset of a vortex motion behind the flow control device.

The present study aims to obtain detailed characteristics of the wake flow behind the plasma wedge actuator to determine its effect on the flow and to compare the results with numerical simulations.

## 2. Arrangement of Experiments

The experiments were performed in a T-325 blowdown wind tunnel, which is located at the Institute of Theoretical and Applied Mechanics (ITAM) in Novosibirsk. The T-325 is an ejector-type wind tunnel with an exhaust into the atmosphere and has the following parameters: range of Mach numbers: 0.5–4; range of unit Reynolds numbers $Re_1 = (3–100) \times 10^6$ (1/m); range of stagnation pressure up to $1.4 \times 10^6$ Pa; dimensions of the working part $0.2 \times 0.2 \times 0.6$ m; and duration of the run up to 60 min. A more detailed description of the T-325 wind tunnel can be found in paper [55].

Experiments were conducted at the free-stream Mach number M = 1.45, stagnation pressure $P_0 = 0.75$ bar, and stagnation temperature $T_0 = 288$ °K, which corresponds to the unit Reynolds number of $Re_1 = 11.5 \times 10^6$ (1/m). The experiment arrangement is illustrated in Figure 3.

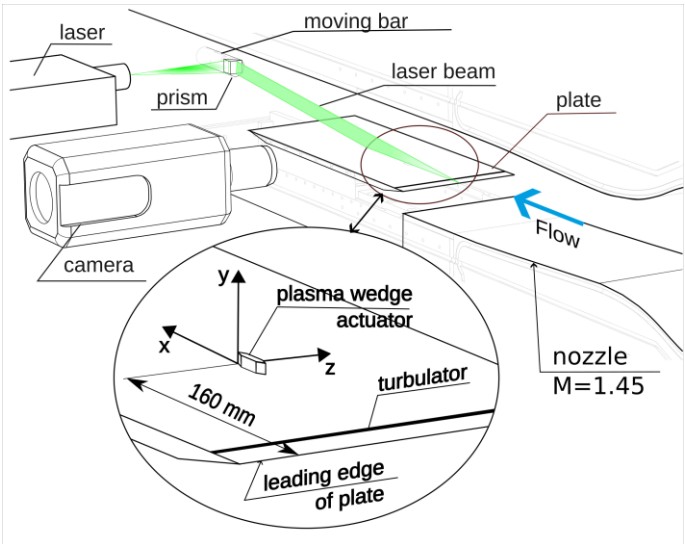

**Figure 3.** Experiment arrangement.

The actuator was fabricated from machinable glass ceramic (MACOR); its photograph is shown in Figure 2. The wedge length (in the $X$ direction) was 13 mm, its width (in the $Z$ direction) was 1.3 mm, and its height (in the $Y$ direction) was 2 mm, which was approximately equal to δ. The centers of the two electrodes made of copper wire 1 mm in diameter were separated by a distance of 6 mm. The actuator was mounted on a model of a flat plate with a sharp leading edge installed at a zero angle of attack. At a distance of 25 mm from the leading edge, a turbulator was installed. The turbulator was a straight-line band with a height of 0.1 mm. The right-hand coordinate system was fitted to the trailing edge of the actuator; thus, the origin of the coordinate system was located at a distance of 160 mm from the leading edge of the plate on the plane of symmetry. The $X$ axis is aligned

in the streamwise direction, and the $Z$ axis is directed to the right. The discharge is located on the right side of the wedge ($Z > 0$).

The flow velocity and the Pitot pressure in the wake behind the actuator were measured to study the characteristics and structure of the flow. The measurements were also performed using a Dantec Dynamics PIV system with a frequency of 15 Hz (laser Litron-Nano L 135-15; Camera Phantom 310 m; Nikkor Lens f105/2.8). The measurement region was in the plane parallel to $XY$ (see Figure 3). The laser beam was introduced into the flow using a prism that was mounted on a moving bar. This allowed measurements to be taken in different planes along the transverse coordinate Z, in the same scale, by moving the prism and the camera. The measurements without and with the discharge were performed in one experiment; more than 1000 pairs of images were obtained for each flow regime. A more detailed description of the features of the PIV method in this wind tunnel can be found in paper [56].

The pressure was measured using a Siemens KPY 43A absolute pressure sensor with a Pitot tube whose outer and inner diameters were 0.7 and ~0.5 mm, respectively. The measurements were performed near the model surface along the spanwise direction $Z$ in several $X$ positions. Additionally, the temperature of the plate surface was measured using the infrared camera FLIR SC7300.

For high-speed flows, realizing a stationary discharge is a very difficult task, because the arc of the electric discharge will not be stable due to the motion of the hot gas by the flow. Due to the external flow, the gas heated in the plasma will move downstream. The heated ionized gas has less resistance and the electric arc will tend to follow the moving zone of the heated flow. As a result, the gas heating zone becomes non-stationary. To avoid this effect, it was decided to use pulsed gas heating, but with a high discharge generation frequency, in order to implement a quasi-stationary gas heating scenario. This allows for a quasi-stationary scenario, in which a new spark discharge occurs before the gas heated by the previous spark discharge has time to leave the spark gap region.

A high-voltage generator was used to excite the spark discharge, enabling the actuator to operate at a frequency of approximately 25 kHz. The energy supply was performed in the immediate vicinity of the walls in the corner region, where the flow velocity significantly decreased. The flow velocity in the plasma channel region was estimated to be within 200 m/s; for this reason, the energy supply was assumed to be quasi-steady. The electrical characteristics were measured using a RIGOL DS1102E oscilloscope through a Tektronix P6015A high-voltage probe and a Tektronix P6021 current probe. The discharge power was calculated by multiplying the measured current and voltage signals and was about 14 W, which corresponds to a specific power of 118 MW/m$^3$. Figure 4 shows examples of voltage and current oscillograms at different time scales.

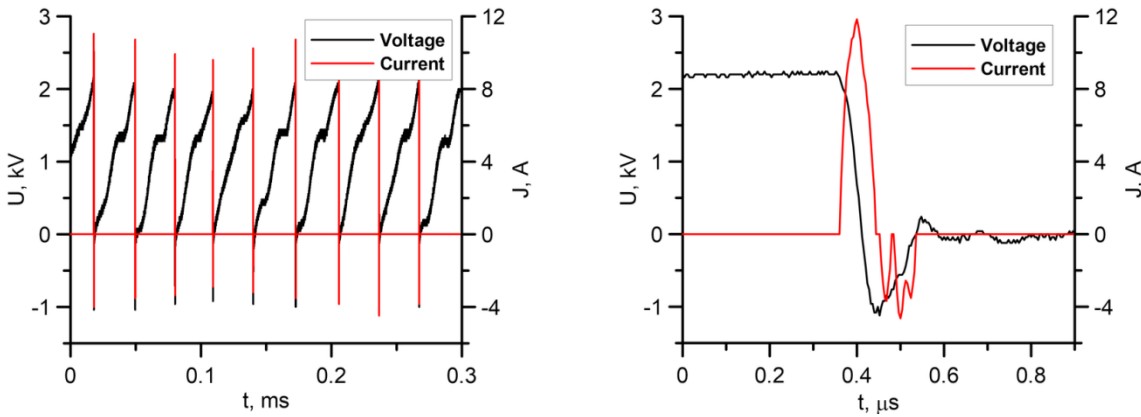

**Figure 4.** Examples of voltage and current oscillograms at different time scales.

## 3. Numerical Setup

Numerical simulations were performed to obtain a better understanding of the physical aspects of the examined phenomenon. The problem was simulated numerically with the aid of the software package ANSYS Fluent. The steady Navier−Stokes equations were solved using a density-based solver. An implicit second-order scheme in space was applied together with the Roe method of splitting the convective fluxes. RANS simulations were performed using the turbulent model $\kappa\text{-}\omega$ *SST*. The capability of conjugate heat transfer calculation is accomplished by integrating a heat conduction procedure in solid bodies (plate and wedge). A structured block grid refined in the wedge region and in the wake behind the wedge was used (Figure 5). Grid refinement in the boundary layer ensured $y^+ \approx 1$ in the entire computational domain. The total number of cells was approximately 7.5 million. The physical size of the computational domain was $205 \times 80 \times 140$ mm, and the wall was modeled by a region 5 mm thick. The characteristics of the thermal conductivity of the wall materials and the size of the wedge were consistent with the real experimental model with some simplifications: internal voids, rounding radii, electrodes, wires, and sealant were not modeled.

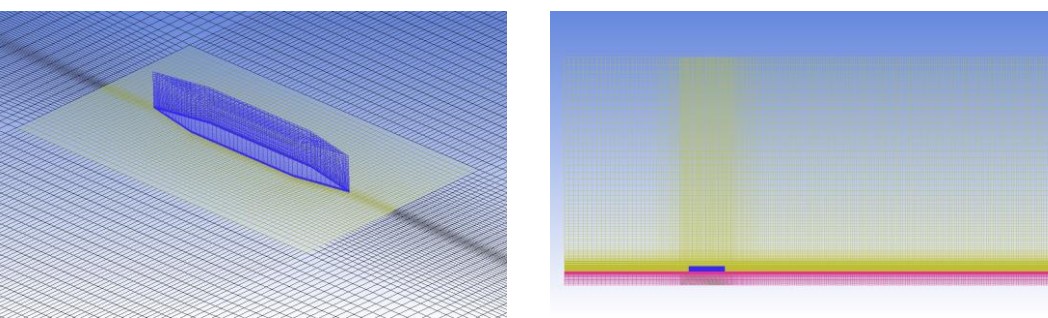

**Figure 5.** Example of mesh in numerical domain.

The thickness of the boundary layer in the region of the actuator was approximately 3 mm. The number of the computational grid cells in the boundary layer was about 55. The value of $y^+$ did not exceed 1. The grid was refined both in the direction of the model wall and in the direction of the wedge. The $x^+$ and $z^+$ values on the wedge surface were about $15 \div 30$. With the distance from the wedge, the $x$ size of the cell increased and the value of $x^+$ was about 50. Grid convergence was tested using the wall shear stress as a criterion. The difference in drag between the basic mesh (7.5 million cells) and a more accurate mesh (45 million cells) did not exceed 1%. When comparing the numerical results with the experiment, calculations were carried out using various turbulent models. The numerical results obtained using the $\kappa\text{-}\omega$ *SST* turbulent model showed the best agreement with the experimental data.

The experimental free-stream parameters were specified at the inlet boundary of the computational domain ($P_0 = 75$ kPa, $T_0 = 290$ K, and M = 1.45). At the outlet boundary, the flow expiration condition was set. The symmetry condition was selected at the upper boundary and side wall of the computational domain. In the calculation, the effects of a real gas, which may arise due to gas heating in an electric discharge, were not taken into account. Viscosity was calculated using the Sutherland equation.

The boundary conditions imposed at the inlet of the computational domain corresponded to a turbulent boundary layer formed on a flat plate at a distance of $X = 100$ mm from the leading edge, which was previously computed in the two-dimensional formulation. The width and height of the computational domain ensured the returning of reflected shock waves formed by the wedge onto the model. The external boundary of the solid model was subjected to adiabatic boundary conditions.

The calculations were performed in two stages. The purpose of the first stage was to confirm the possibility of numerical simulation of the problem under study with the

simplifications used (RANS model, etc.). The purpose of the second stage was to study the process of streamwise vortex generation.

To achieve the goals of the first stage, it was necessary to accurately reproduce the real experimental geometry of the wedge. Exact measurements of the fabricated wedge revealed some differences from the design (Figure 6). Therefore, it was decided to calculate both the design and real geometry. It should be noted that only the inaccuracies of upper surface fabrication were taken into account in real wedge calculations. The wedge walls were assumed to be perpendicular to the model, which was probably not the case in reality. Moreover, minor spalls on the frontal and rear parts of the wedge were ignored.

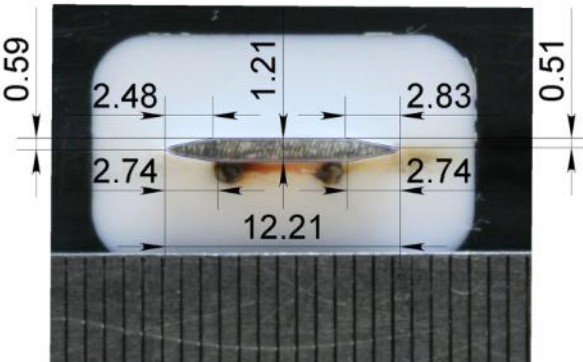

**Figure 6.** Sizes of the fabricated wedge.

To solve the problems of the second stage, it was decided to investigate the effect of the discharge location. Four cases were considered, which are shown in Figure 7. The first case LE corresponds to the location of the heating area near the leading edge; in the second and third cases, SP discharge is located in the middle of the wedge, and it differs in length. The fourth case TE corresponds to the discharge located near the trailing edge. The integral value of the power input to the heating zones was 5.5 W for most cases. Only for the SP case with a long heating area was the amount of energy emitted 10 W.

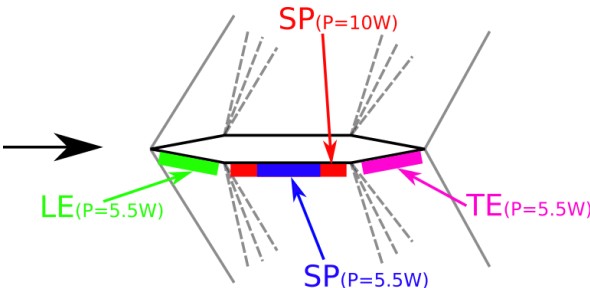

**Figure 7.** Locations of zones of energy supply near the wedge in the calculations.

## 4. Experimental Results

The velocity profiles measured by PIV with energy supply ($N$ = 14 W) and without energy supply ($N$ = 0) are compared in Figure 8. The velocity profiles for the stagnation pressure $P_0$ = 0.75 bar are presented in the logarithmic scale in the two most typical cross-sections along the Z axis. The velocity is normalized to the free-stream velocity $Ue$, and the vertical coordinate $Y$ is normalized to the undisturbed BL thickness $\delta_0$, which was taken as the BL thickness in the cross-section $Z$ = 8 mm for the corresponding value of $X$. The thickness of the boundary layer was estimated from PIV measurements. The point at which the flow velocity was equal to 99% of the oncoming flow velocity was taken as the edge of the boundary layer. All velocity profiles correspond to the turbulent profile, and the deviation from the linear dependence is associated with the influence of the wedge on the flow. It is seen in Figure 8a corresponding to $Z$ = −2 mm that the velocity profile becomes

more filled in the case of energy supply for all values of *X*. In turn, on the other side of the wedge, at *Z* = 2 mm (Figure 8b), the velocity profile in the case with energy supply becomes less filled (again, for all values of *X*). Comparing Figure 8a,b, one can see that the BL thickness for *Z* = 2 mm is greater than that for *Z* = −2 mm.

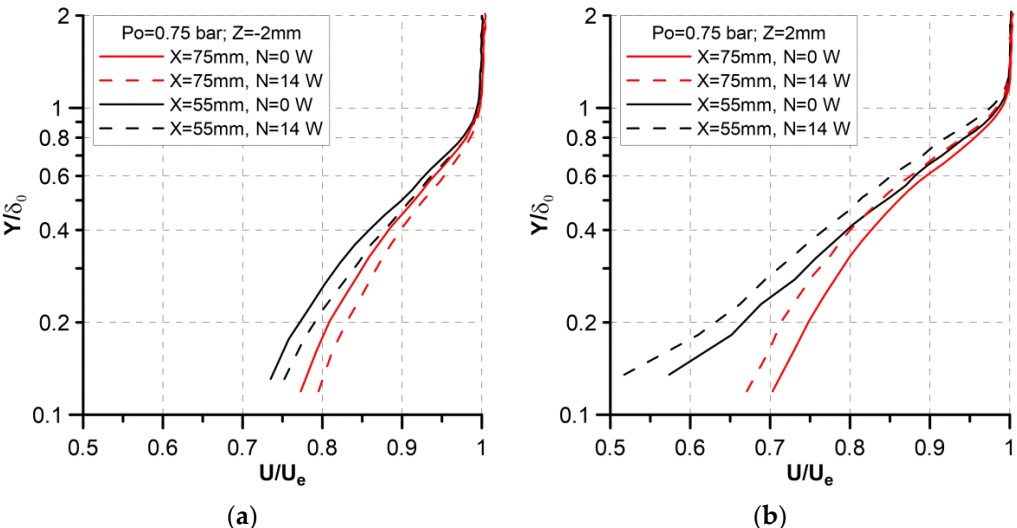

**Figure 8.** Velocity profiles at the logarithmic scale (**a**) *Z* = −2 mm, (**b**) *Z* = 2 mm.

The evolution of the velocity distribution in the boundary layer can be traced in more detail by considering the velocity fields in the *YZ* plane shown in Figure 9. These distributions were obtained by combining PIV measurements performed in several *Z* cross-sections from −3 to 2 mm for *X* = 75 mm. The right and left pictures show the flow fields without and with energy supply, respectively. Deformation of the velocity field in the plane of symmetry *Z* = 0 is clearly visible. The BL thickness is seen to decrease for *Z* < 0, whereas a small increase in δ is observed for *Z* > 0. Thus, the flow in the wake behind the actuator is originally asymmetric, and the effect of velocity field deformation is enhanced on both sides of the plane of symmetry when the discharge is switched on. In particular, for *Z* < 0, the BL thickness in the case of energy supply decreases more significantly than in the basic case; for *Z* > 0, the BL thickness increases to a greater extent in the case with the discharge than in the case without external effects. Based on these data, it can be concluded that the actuator forms a vortex in the wake in both the active and passive regimes. Moreover, energy supply leads to the desired effect and increases the vortex intensity. It should be noted that vortex formation in the passive mode is unexpected and contradicts the results of preliminary numerical simulations. An apparent reason is wedge asymmetry caused by fabrication inaccuracies. To explain this result, we performed additional computations; the results are discussed below.

For a more illustrative presentation of the velocity field evolution and vortex intensification effect, it is convenient to construct the field of the velocity difference in the cases with and without energy supply. Such fields of the velocity difference (defect) are shown in Figure 10. From the figure, one can clearly see the region on the right of *Z* = 0, where the velocity decreases owing to the discharge; on the left of the zero point, there is a region where the discharge induces an increase in velocity. In both cases, the region of velocity reduction is greater than the region where the velocity increases. Thus, the overall effect induced by the discharge is the decrease in the momentum in the boundary layer. However, the main region of velocity reduction is located at a greater distance from the surface than the region where the velocity increases, which almost reaches the surface. The asymmetric behavior of velocity also confirms vortex formation and its intensification in our case.

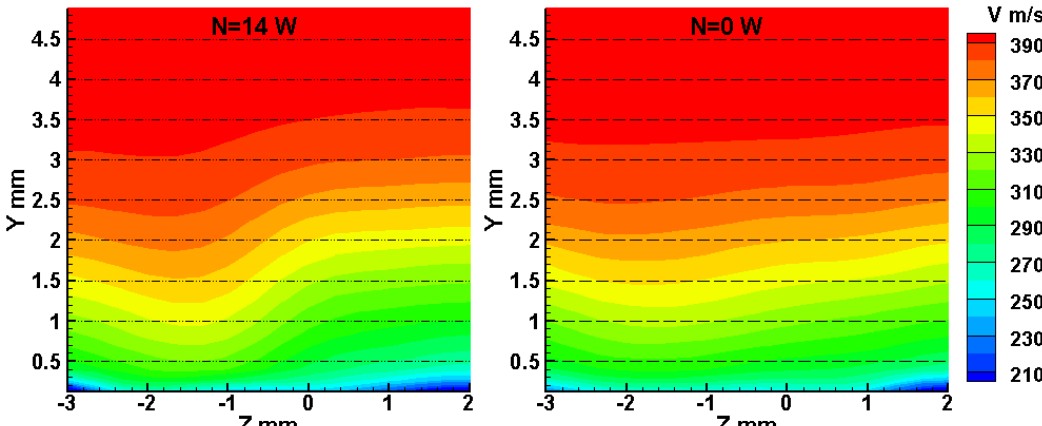

**Figure 9.** Fields of the velocity for *X* = 75 mm.

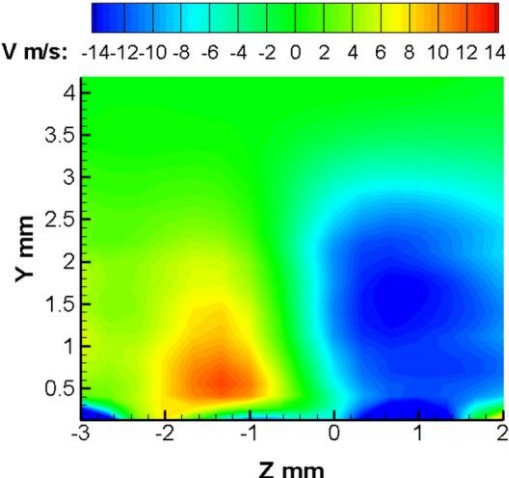

**Figure 10.** Fields of the velocity defect for *X* = 75 mm.

Figure 11 shows the Mach number and static temperature distributions over the dimensionless coordinate $Y/\delta_0$ as compared to the Pitot tube size. The miniature Pitot tube was located along the flow and mounted on a movable bar instead of a prism, which was used in the PIV experiments. The static temperature values are calculated via the Crocco integral with the use of data for the velocity profile in the cross-section *Z* = 8 mm, *X* = 55 mm, which is close to the coordinate where the Pitot measurements were performed and the data for the total pressure and temperature in the settling chamber in the corresponding experiment. The figure shows that the Pitot tube, which has the outer diameter of 0.7 mm, is completely located in the subsonic region of the boundary layer. This means that there are no losses on the shock wave; hence, it is possible to perform direct comparisons of pressure over the entire wake width. Figure 12 shows the ratio of the pressure measured by the Pitot tube to the total pressure. An increase in pressure on one side of the axis of symmetry and its decrease on the other side of the axis of symmetry testifies that the actuator generates a vortex in both passive and active regimes. This observation is consistent with the PIV data presented above. It is seen that the change in pressure is more significant in the case with energy supply. It can be noted that the deformation of the pressure distribution decreases in the downstream direction, and the difference between the cases without and with energy supply increases.

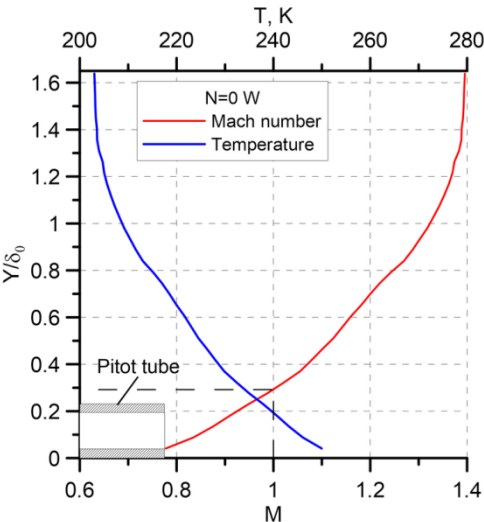

**Figure 11.** Comparison of the Pitot tube size with the Mach number and static temperature distributions over the vertical coordinate.

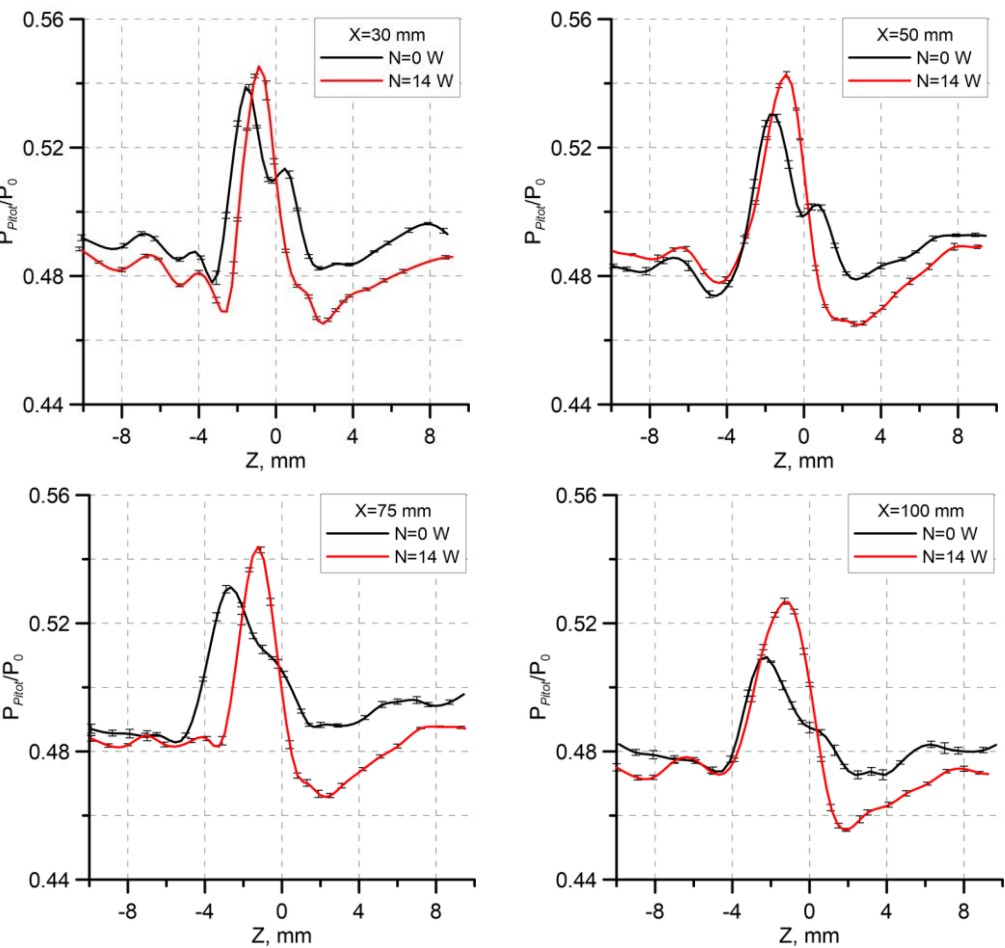

**Figure 12.** Ratio of the pressure measured by the Pitot tube to the total pressure in the incoming flow.

Figure 13 shows the curves of the pressure measured by the Pitot tube and the momentum thickness calculated on the basis of PIV data. For more convenient comparisons, the pressure axis is turned upside down because the total pressure growth in the boundary layer is caused by the fact that the profile becomes more filled and, hence, the displacement thickness decreases. It should also be noted that the momentum thickness was calculated

using the formula for incompressible flows for simplicity, as only qualitative comparisons were important. It is seen that the curves are similar to each other. Both quantities display a change (decrease) in the curve slope in the region $Z = 0$ for the case without the discharge ($N = 0$ W).

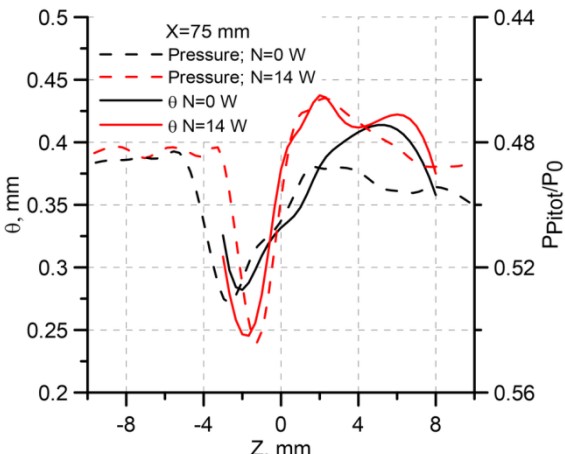

**Figure 13.** Displacement thickness and pressure versus $Z$.

## 5. Comparison of Experiment and Numerical Results

The plasma was simulated as a steady 3D region of energy supply. The choice of a steady energy supply as a discharge model was made to save computational resources and time. This choice is justified by the fact that the main purpose of this actuator is to influence the average (in time) characteristics of the flow, which is possible only by a steady or "quasi-steady" influence as in the experiment. In this case, the shock waves or ion wind created by the discharge are too weak to noticeably affect the flow, so only the thermal effect is taken into account. The energy was supplied in the middle of the wedge on its right side (in the streamwise direction) on a straight-line segment. The plasma region boundaries touched the wedge wall and the model. The energy supply region size was $5 \times 0.5 \times 0.5$ mm. The computations were performed for three values of the energy density of the heating region (20, 40, and 80 MW/m$^3$), which corresponded to the discharge power $N = 2.5$, 5, and 10 W.

The calculated Pitot pressures are shown in Figure 14. The Pitot pressure was calculated by means of averaging the stagnation pressure in the region corresponding to the Pitot tube cross-section in the experiment. When there is no discharge, a weak asymmetric distortion of the flow is formed behind the design wedge. This is caused by the formation of two weak streamwise vortices behind the wedge, which gradually decay in the wake region. The asymmetry in the real wedge geometry leads to the formation of one powerful streamwise vortex (Figure 14b). This means that the chosen wedge structure is extremely sensitive to fabrication inaccuracies and should be improved, e.g., by changing the geometry of the leading and trailing edges. Comparing Figures 12 and 14b for the case without the discharge, one can see that the data are in good agreement, especially in the maximum pressure region. Possible reasons for the differences in the minimum pressure region are the inaccuracy of retrieval of the real wedge geometry and the errors of RANS modeling of the regions of shock wave–boundary layer interaction.

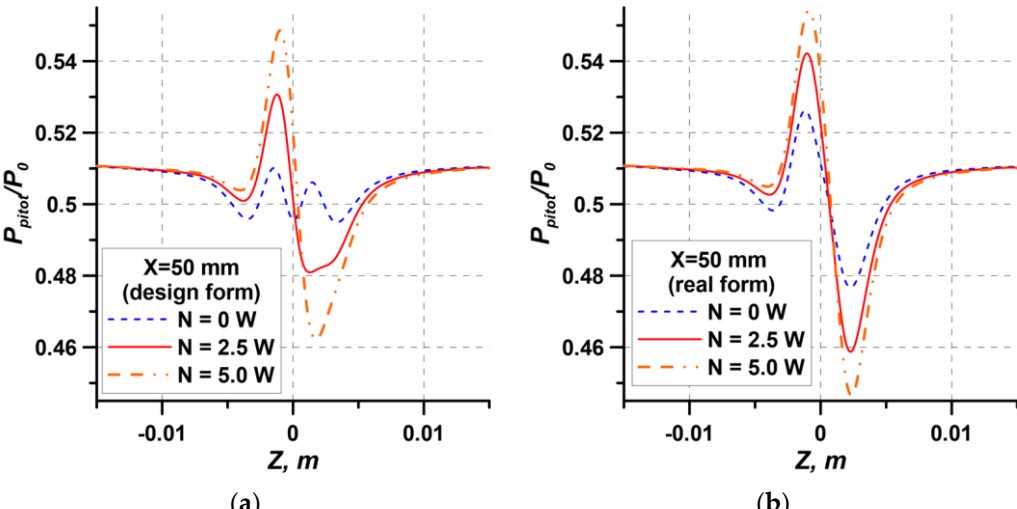

**Figure 14.** CFD-based distributions of stagnation pressure in the boundary layer behind the (**a**) design and (**b**) real wedge in the cross-section $X = 50$ mm.

Discharge actuation leads to the formation of a streamwise vortex for the designed geometry and vortex intensification for the real geometry. It is seen from the data obtained that a change in the discharge power in the case of design geometry leads to a greater change in pressure than in the case of real geometry. This means that the presence of the initial vortex motion slightly decreases the efficiency of streamwise vortex generation by using energy supply. The increase in the calculated maximum stagnation pressure due to plasma actuation, as compared to the experimental data, was also observed for a lower discharge power. Possible reasons are the inaccuracies of modeling the energy supply region (exact data on the shape and temperature distribution in the plasma channel are not available), errors of wedge geometry retrieval, and imperfections of the RANS turbulence model. Nevertheless, the data obtained allow us to argue that the computations ensure correct simulation of the qualitative behavior of the plasma wedge effect and the thermal action of the discharge is the governing factor. Correspondingly, numerical simulations can be used to refine the physical issues of interaction of the energy supply region with the wedge and to optimize the shapes of the wedge and energy supply region.

In the experiments, the discharge power was calculated on the basis of the measured current and voltage oscillograms. Unfortunately, these measurements do not take into account the losses in conductors and electrodes. As the discharge power affects the wall temperature distribution, it was decided to compare the calculated and experimental data. The results in Figure 15 are obviously in good qualitative agreement. The thermal wake width in the experiment is slightly greater than the predicted value, which can be caused by the insufficient resolution of the IR camera used, which provided no more than 30 pixels in the $Z$ direction. With allowance for the spatial resolution of the IR camera, it can be argued that the experimental temperature distribution corresponds to 5 W of the discharge power in the computations. The reason for the difference between the experimental and calculated power may be due to the error in the measuring of the electrical parameters and the fact that part of the electrical power is spent on heating conductors and capacitors. Figure 15b shows the data calculated for the real wedge, but the difference in the wall temperature distributions for the real and design geometry was negligible.

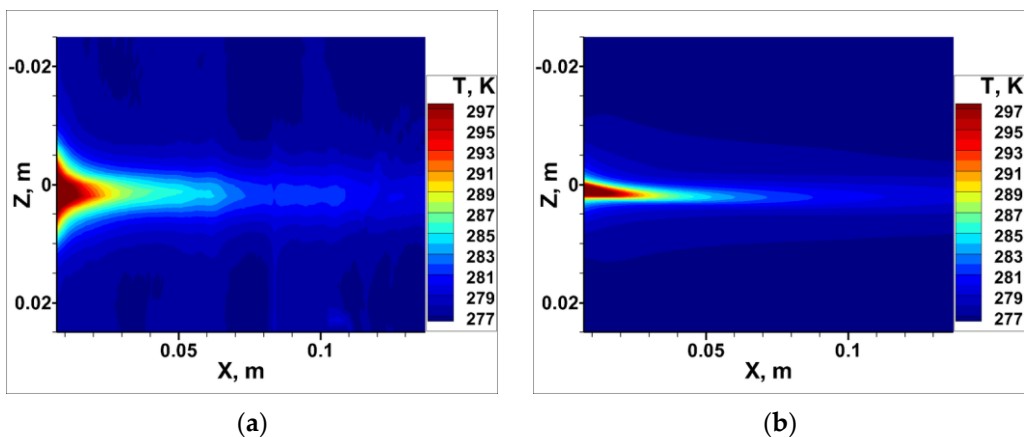

**Figure 15.** Temperature distribution on the model wall: (**a**)—experiment, *N* = 14 W; (**b**)—computations, *N* = 5 W.

The discharge effect on the streamwise vortex evolution for the design and real geometry is demonstrated by an example of the shear stress distributions on the model wall (Figure 16). As the discharge power increases, the shear stress gradually increases in the region of negative *Z* and decreases on the opposite side (where plasma occurs). Further downstream, the peak values of the shear stress decrease, and the width of the vortex region increases less intensely, corresponding to the gradual decay of vortex motion. For the real geometry, the vortex rotation direction coincides with the direction generated by the plasma action. Therefore, discharge actuation leads to intensification of the existing vortex, similar to the experiment.

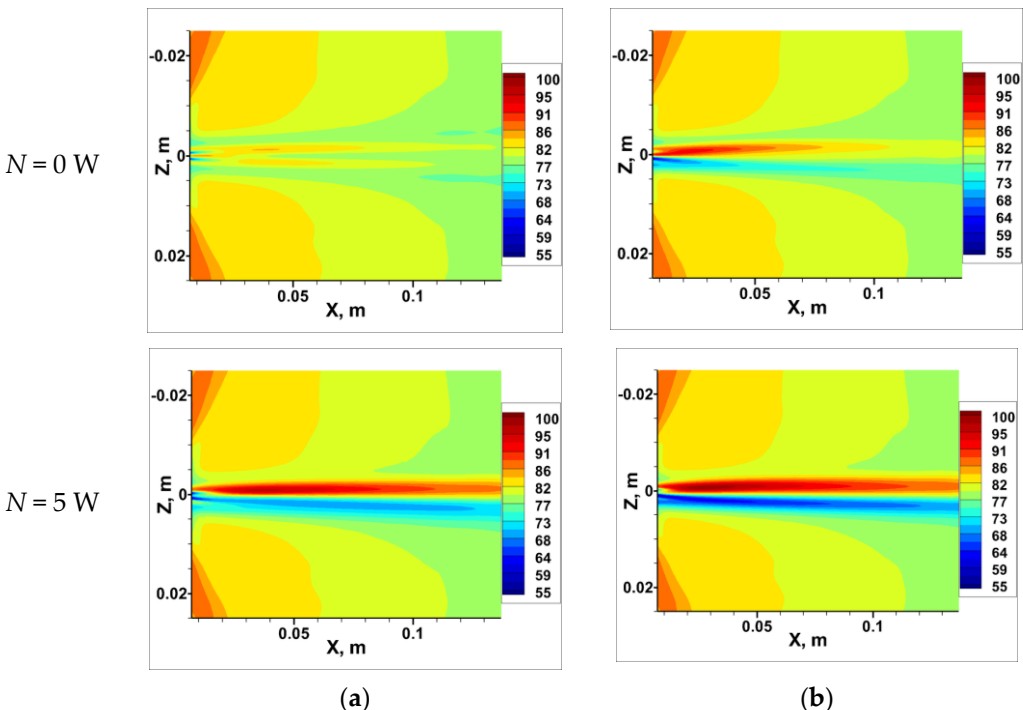

**Figure 16.** CFD-based shear stress distributions [Pa] on the model wall behind (**a**) design and (**b**) real wedges.

As a result of comparing experiments with numerical data, it was found that the effect of electric discharge on the flow can be modeled by the volumetric heat release in the RANS method with good accuracy.

## 6. Data Processing of Numerical Results

It was decided to conduct additional numerical studies to enhance the understanding of the streamwise vortex formation mechanism by the electric discharge located near the fin. The aim of these studies was to investigate the impact of the location of the energy deposition region. Therefore, the flow downstream the fin without discharge is considered as the basic flow.

The role of the turbulent boundary layer in the formation of the vortex was investigated separately. In order to simulate the absence of a boundary layer, slip boundary conditions were set on all channel walls. The numerical problem was solved using RANS equations with the turbulence model to improve the convergence of the solution. Figure 7 shows four variants of the location of the energy deposition region.

Figure 17 shows the distribution of the displacement thickness defect downstream of the wedge. To calculate this value, we subtracted the distribution of $\delta^*$ of the base flow from the corresponding value of the flow with active discharge. Qualitatively, all test cases are similar, except for Figure 17a, which corresponds to the location of the discharge near the leading edge of the fin. All other figures clearly show that the excitation of the discharge results in two regions of negative and positive displacement thickness defect, which corresponds to the formation of a streamwise vortex. The maximum effect is observed for the test case with maximum energy release (Figure 17c). With the same energy release and the location of the discharge near the trailing edge or along the middle part of the wedge (Figure 17b,d), there is no significant difference in the results. When the discharge is located near the leading edge, the main effect on the flow is observed near the wedge. Downstream, the displacement thickness defect tends to zero. This means that the vortex is practically not formed.

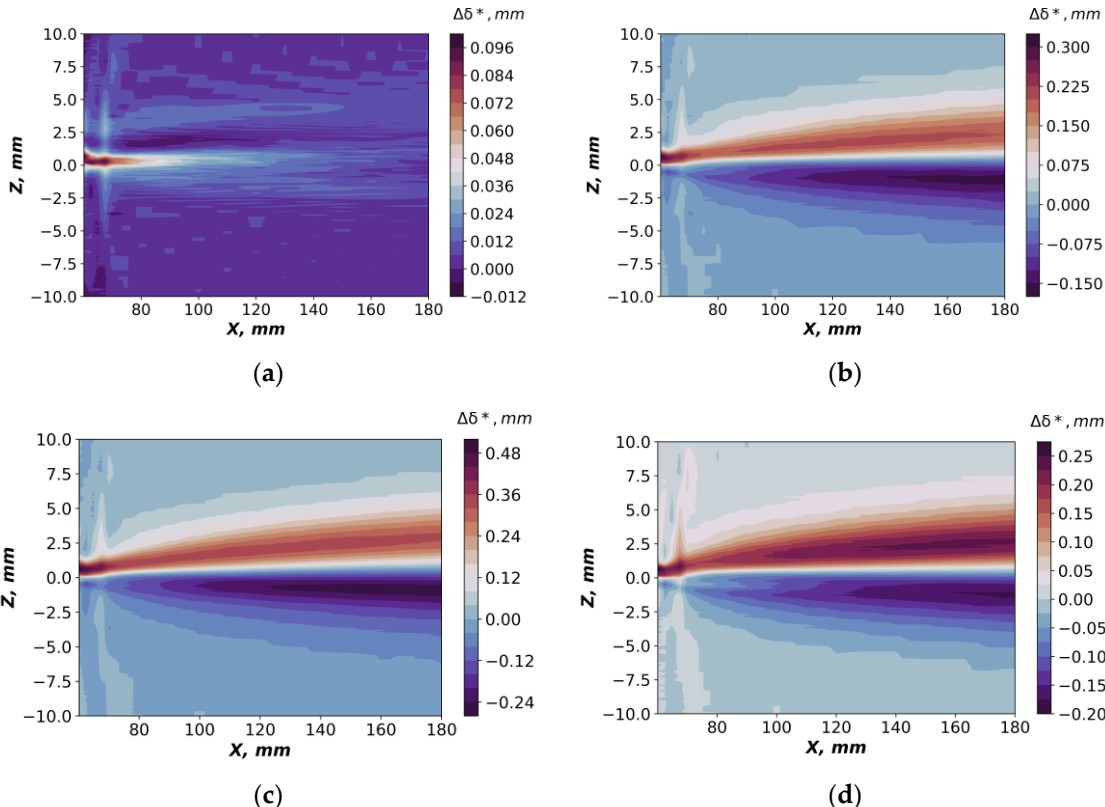

**Figure 17.** Defect of displacement thickness: (**a**) Case LE, P = 5.5 W, (**b**) Case SP, P = 5.5 W, (**c**) Case SP, P = 10 W, and (**d**) Case TE P = 5.5 W.

Figure 18 shows the variation in the minimum and maximum of the momentum thickness defect along the minimum/maximum line as it moves downstream. It is clearly

noticeable that for the same value of the discharge power, the momentum thickness defect depends little on the location of the discharge (except for the discharge near the leading edge). It can be seen that the magnitude of the momentum thickness defect increases downstream. This means that the streamwise vortex does not dissipate significantly and continues to contribute to the momentum exchange. The calculations also evaluated the effect of the vortex on the stagnation pressure defect. It was found that the increase in the flow power caused by the total pressure defect approximated the discharge power. The increase in the flow power was calculated by the below formula:

$$\Delta N = U_e \cdot \iint \left( P_0^{dis} - P_0^{ref} \right) dy \, dz \qquad (1)$$

where $P_0^{dis}$—stagnation pressure for the case with plasma, $P_0^{ref}$—stagnation pressure for the case without plasma, and $U_e$—incoming flow velocity.

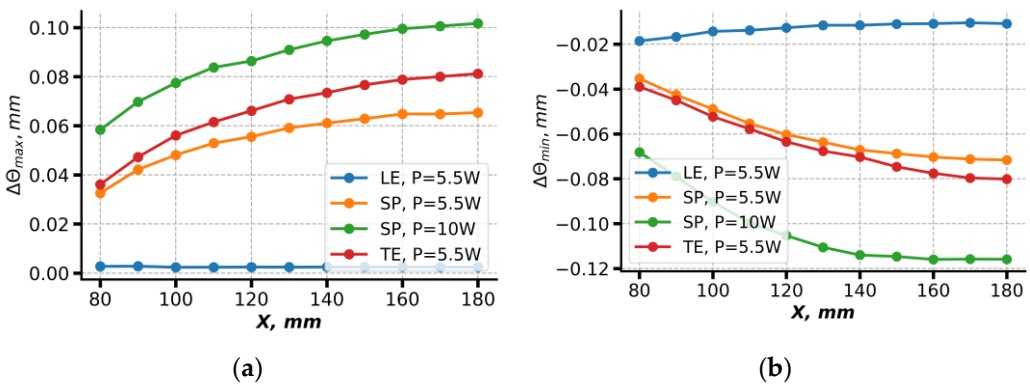

(a)  (b)

**Figure 18.** Distribution of the maximum (**a**) and minimum (**b**) defects of momentum thickness along the streamwise coordinate.

Let us consider the flow field formed immediately downstream of the actuator. Figure 19 shows the distributions of the magnitude defect. On the left, the data are shown for the case of the discharge location near the leading edge, LE, and on the right, for the middle location of the discharge SP (Figure 7). The upper figures show the defect of the streamwise velocity component, and the lower figures show the defect of the static pressure. In addition, the upper figures show streamlines, and the lower figures show the distribution of the static temperature defect by isolines. The green dot in all figures corresponds to the position of the static pressure minimum. The data in Figure 19 were obtained for the test case with a boundary layer. Similar data, but for the case without the boundary layer, are shown in Figure 20. Recall that the energy input region is at the positive transversal coordinate.

Let us first consider the test case without a boundary layer (Figure 20). Comparing the test cases LE and SP, we can see that the distributions of the pressure and temperature defect are qualitatively almost the same. Basically, the increase in pressure is observed directly behind the discharge. The maximum temperature defect is also observed in that region. The value of the maximum temperature defect in this section was approximately 130 degrees for the LE case (Figure 20c) and 220 degrees for the SP case (Figure 20d). At the same time, the velocity fields differ more significantly. For the LE case (Figure 19a), a region of increased velocity is formed in the discharge wake, whereas in the SP case, the velocity decreases in the wake. It was shown in [52] that volumetric heating of the gas in the discharge channel leads to an increase in velocity in the heating region. The heating of the gas also leads to an increase in the speed of sound, resulting in a decrease in the local Mach number. When the heating region is located near the leading edge, the heated gas passes through two expansion fans, whereas for the case of heating in the middle part of the wedge, it undergoes acceleration only once. As a result, for the LE case, the Mach number just upstream of the trailing edge shock wave is significantly higher than for the SP

case. Therefore, an oblique shock wave is formed near the trailing edge in the former case, whereas a normal one is formed in the latter case and a significant decrease in the heated gas velocity is observed in the SP case. In spite of a noticeable pressure drop on different sides of the wedge, no obvious vortex formation is observed in its immediate vicinity for both the LE and SP cases (see the streamlines in Figure 20).

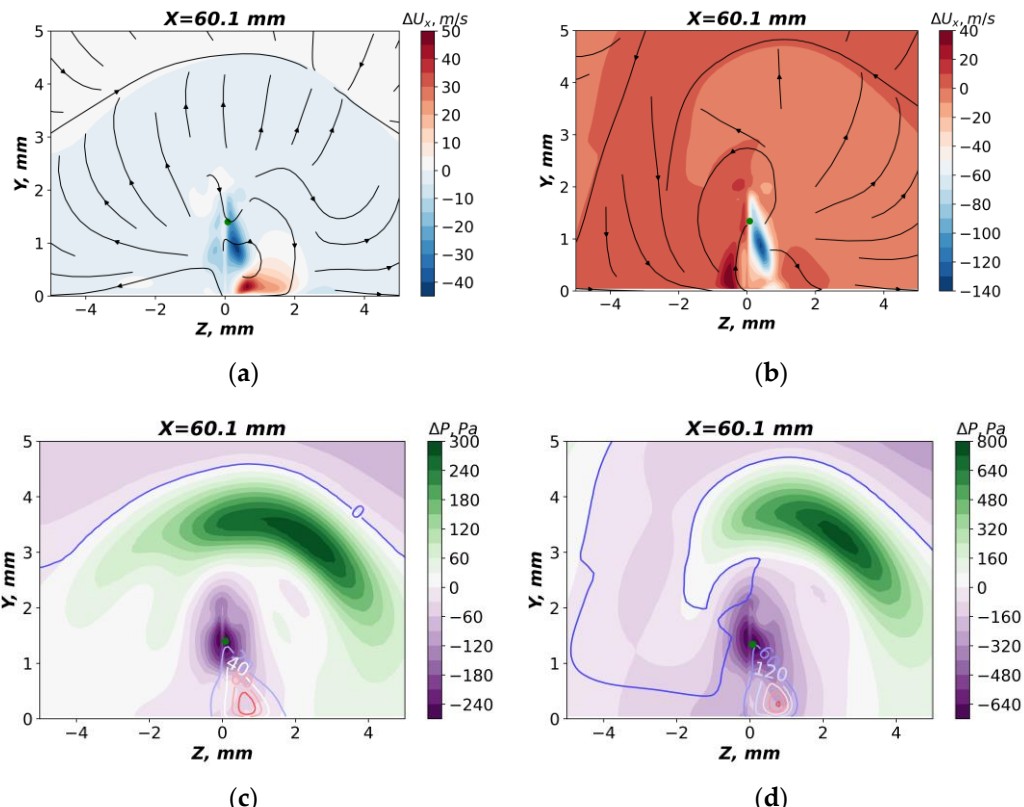

**Figure 19. Top**—distribution of the defect of streamwise velocity; **bottom**—distribution of the defect of pressure and isolines of the temperature defect at X = 60.1 mm ((**a,c**)—Case LE, P = 5.5 W; (**b,d**)—Case SP, P = 10 W).

The appearance of the boundary layer leads to a significant change in the flow pattern (Figure 19). The presence of the boundary layer leads to an increase in the size of the positive temperature defect region. In this case, the pressure defect in the region of heated gas becomes close to zero. The main pressure rise is now observed above the boundary layer on the discharge side. This pressure rise may be explained by the presence of an oblique shock wave originating from the region of heated gas. It may also be noted that in the case of the boundary layer, there is a significant defect in velocity and pressure observed on both sides of the wedge, in contrast to the case without the boundary layer. This can be explained by the possibility of perturbations propagating against the flow and around the wedge, due to the presence of subsonic zones in the near-wall layer. It is interesting to note that the velocity defect obtained for the case with a boundary layer is similar to that of the case without a boundary layer. For the SP case, the velocity defect in the wake is mostly negative, and for the LE case near the wall a positive velocity defect is observed. For the SP case, the streamlines show the appearance of swirling motion in the wake downstream from the wedge, which can be interpreted as the formation of a streamwise vortex. For the LE case, no such fluid motion is clearly detected.

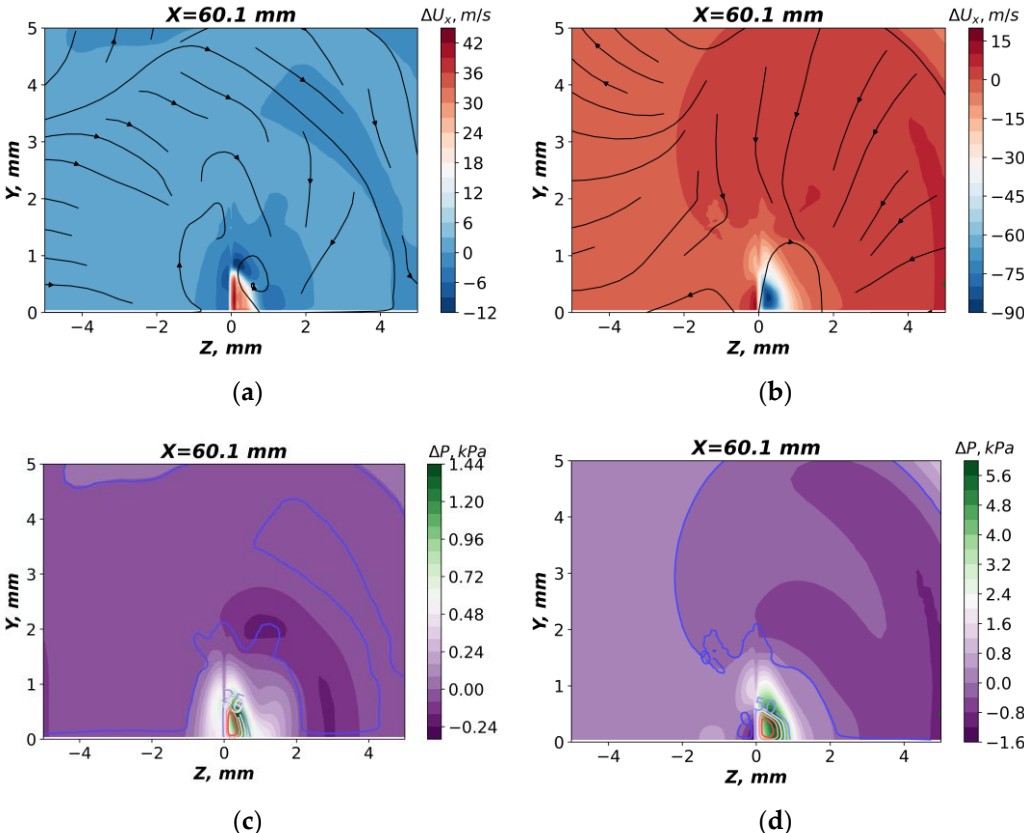

**Figure 20. Top**—distribution of the defect of streamwise velocity; **bottom**—distribution of the defect of pressure and isolines of the defect of temperature at X = 60.1 mm for case without BL ((**a**,**c**)—Case LE, P = 5.5 W; (**b**,**d**)—Case SP, P = 10 W).

Figures 21 and 22 show the flow parameters in the wake at a significant distance from the wedge (ΔL = 70 mm). For the boundary layer case, it corresponds to ΔL/δ* = 118, where δ* is the displacement thickness of the boundary layer just upstream of the wedge. Figure 21 clearly shows the formation of the streamwise vortex in the presence of the boundary layer. The vortex forms regions of high and low velocity defect in the boundary layer, increasing the momentum exchange. For the SP case, the vortex intensity is significantly higher than for the LE case, which can be seen by the higher velocity defect. The point of pressure minimum in Figure 21b (SP case) coincides well with the rotation center of the flow, which confirms the presence of the streamwise vortex. For the LE case, the vortex intensity is significantly less, so it does not form the center of the pressure minimum. Nevertheless, it can be noted that the streamwise vortex in the far wake occurs for all of the test cases. In addition, the heat wake widens downstream and the temperature maximum is located near the wall.

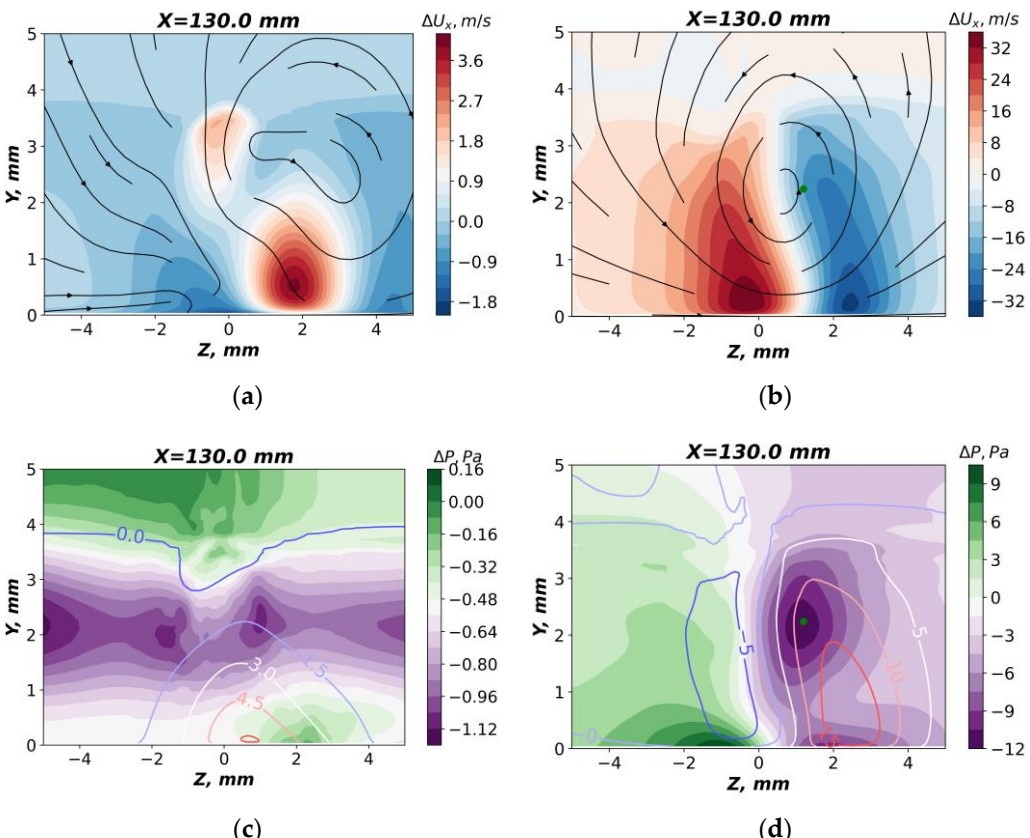

**Figure 21. Top**—distribution of the defect of streamwise velocity; **bottom**—distribution of the defect of pressure and isolines of the defect of temperature at X = 130 mm ((**a**,**c**)—Case LE, P = 5.5 W; (**b**,**d**)—Case SP, P = 10 W).

A more complicated flow in the far wake was obtained for the test cases without a boundary layer (Figure 22). Regardless of the region of energy deposition, an increased velocity is observed in the wake behind the wedge, which most likely arises due to displacement of the flow by the discharge. For the LE case, no significant flow distortion is observed except for the region of the flow acceleration. For the SP case, a well-detectable streamwise vortex with a center at $Z_c \approx 1.5$ mm, $Y_c \approx 0.9$ mm, appears. A significant static pressure drop is observed in the center of the vortex. It is interesting to note that the region of positive temperature defect is observed in the vortex.

It is well seen from the data obtained that the streamwise vortex for the SP case occurs both in the presence and absence of the boundary layer. Note that this is also true for the TE case. For the LE case, there is no vortex of significant intensity in the wake. Thus, we can assume an inviscid mechanism of the vortex formation, which is significantly affected by the presence of the boundary layer. In addition, it should be noted that the formation of a vortex is not clearly observed immediately behind the wedge. It can be assumed that the vortex arises gradually downstream due to the flow defects formed near the wedge, under the influence of the electric discharge. This hypothesis is confirmed by the data obtained in Figure 18.

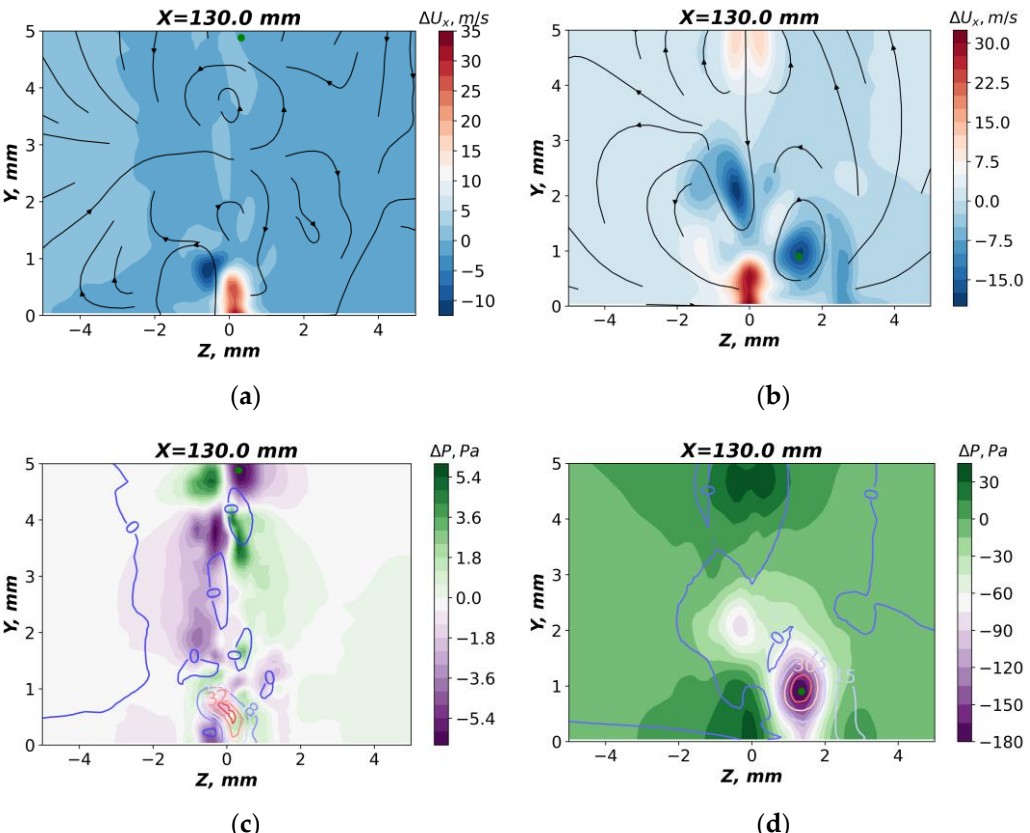

**Figure 22. Top**—distribution of the defect of streamwise velocity; **bottom**—distribution of the defect of pressure and isolines of the defect of temperature at X = 130 mm for case without BL ((**a**,**c**)—Case LE, P = 5.5 W; (**b**,**d**)—Case SP, P = 10 W).

## 7. Conclusions

The present study provided experimental data on the velocity distributions in the wake behind a plasma wedge actuator, pressure distributions along the transverse coordinate, and temperature field on the model surface. In addition to flow parameters, the electric characteristics of the discharge were also measured. The flow behind the actuator was also numerically simulated using CFD methods, and the results were compared with the experimental measurements.

The experimental data showed that the actuator generated a vortex in both active and passive regimes. Vortex generation by the actuator in the passive state was caused by the asymmetry of its shape. Nevertheless, discharge actuation resulted in significant intensification of the vortex, which proved the operability of actuators of this type.

Numerical simulations performed for the real actuator shape showed that the examined flow was rather sensitive to minor asymmetry of the wedge, which led to vortex formation in the wake behind the wedge. Computations of the flow with energy supplied on one side of the actuator were found to be in qualitative agreement with the experimental data, which suggests that the initial idea was correct and that further investigations of this method of flow control can offer significant prospects. The predicted Pitot pressure distribution agrees well with that measured in the experiment, which testifies to the validity of the simplified method of modeling the discharge effect used in the present computations. This conclusion implies that RANS simulations can be used for optimization of the geometry and electric characteristics of the actuator.

A numerical study of the vortex formation process was performed. An effect of the energy deposition location on the wedge was considered as well as the role of the velocity gradient in the boundary layer. It was found that energy deposition near the leading edge does not produce significant vortex in the wake. For the case of flow heating in the middle

and near the trailing edge of the wedge, it is possible to achieve generation of a streamwise vortex, both with and without a boundary layer. Analysis of the flow structure suggests that the streamwise vortex does not occur near the wedge, but downstream due to flow defects formed near the wedge.

**Author Contributions:** Conceptualization, P.P. and A.S.; Methodology, P.P. and O.V.; Formal analysis, P.P. and A.S.; Investigation, O.V.; Data curation, P.P. and O.V.; Writing—original draft, P.P., O.V. and A.S.; Writing—review & editing, A.S. All authors have read and agreed to the published version of the manuscript.

**Funding:** The research was carried out within the state assignment of Ministry of Science and Higher Education of the Russian Federation.

**Data Availability Statement:** The data can be provided on request.

**Acknowledgments:** The study was conducted at the Equipment Sharing Center «Mechanics» of ITAM SB RAS.

**Conflicts of Interest:** The authors declare no conflict of interest.

## Abbreviations

**Nomenclature**

| | |
|---|---|
| M | Mach number |
| $N$ | electric power of discharge [W] |
| $P$ | pressure [Pa] |
| $Re$ | Reynolds number |
| $Re_1$ | unit Reynolds number [1/m] |
| $T$ | temperature [K] |
| $t$ | time [s] |
| $X$ | longitudinal coordinate from rear edge of actuator [mm] |
| $Y$ | vertical position perpendicularly free-stream direction [mm] |
| $Z$ | spanwise direction [mm] |
| $U$ | velocity [m/s] |
| $U/U_e$ | relation of velocity in boundary layer to velocity of the edge of boundary layer |

**Greek**

| | |
|---|---|
| $\delta$ | boundary layer thickness [mm] |
| $\delta_0$ | undisturbed boundary layer thickness [mm] |
| $\delta^*$ | displacement thickness |
| $\theta$ | displacement thickness [mm] |

**Subscripts**

| | |
|---|---|
| $\infty$ | free-stream parameters |
| 0 | stagnation parameters |
| e | parameters of the edge of boundary layer |

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
