# Peer review of "Study of Plasma-Based Vortex Generator in Supersonic Turbulent Boundary Layer"

_aerospace, doi:10.3390/aerospace10040363_

Round 1

Reviewer 1 Report

ABSTRACT:

1-    Since flow control can be used for many applications including wind turbines, airplanes, missiles, etc, I suggest that authors change the word “vehicle” to “aerodynamic object”.

2-    The authors should used consistent format for the numbers “Re=1.5x106

3-    Reynolds in a non-dimensional number. Why is the dimension of “ Re “ [1/m]?

INTRODUCTION:

Overall, I believe that the INTRODUCTION part should be improved. The authors start with vortex generators which seems reasonable and then provide some information about other methods such as SJAs. In my opinion, it is worth talking about the applications of flow control methods/devices. Hence, I suggest that authors refer to some new references about using flow control methods for different applications. I found the following papers can add value to this research:

Ø  Drag Reduction of a Passenger Car Using Flow Control Techniques. Int.J Automot. Technol. 20, 397–410 (2019). https://doi.org/10.1007/s12239-019-0039-2

Ø  Design and aerodynamic performance analysis of a finite span double-split S809 configuration for passive flow control in wind turbines and comparison with single-split geometries, Journal of Wind Engineering and Industrial Aerodynamics 214, 104654

Ø  CFD analysis of a new active flow control method for wind turbine airfoils using a paddle wheel inside the split of airfoil, J. Comput. Fluids Eng. 26 (2), 69-76

Ø  Aerodynamic and Structural Design of a 2022 Formula One Front Wing Assembly. Fluids 2020, 5, 237. https://doi.org/10.3390/fluids5040237

Ø  Hypersonic Flow Control of Kinzhal Missile via Off-Axis, Pulsed Energy Deposition Partth A. Laad et al., AIAA Journals, 2021

Ø  Magnetohydrodynamic Control of Hypersonic Separation Flows, International Journal of Aerospace Engineering, vol. 2021, Article ID 6652795, 13 pages, 2021.

Also, please consider modifications of the following points:

Line 28:

The references [1-3] are too old. I believe that the authors do not need to go back to the history of flow control. In case they want to mention the importance of flow control, they can simply refer to a recent review paper.

Line 35:

There is little value in listing 5 references [5,6,…,9] just by their number. For example, I suggest authors can only select two papers and provide some highlights and important findings of each of them and explain them briefly. The same comment applies to LINEs  64 , LINE 75 and LINE 91

Line 54:

I suggest that authors break the existing paragraph and start a new one from “ A device implementing….”

 SECTION 2

LINE 138:

Please provide the location /owner (laboratory name) of the wind tunnel. Also please list the information of the wind tunnel in a separate Table.

Figure 3:

Is fade. Please replace it with a more clear figure.

 SECTION 3

Line 192:

SST-k-ω is very sensitive to Y+, expansion ratio, and number of cells inside the boundary layer. (Effects of near-wall grid spacing on SST-K-ω model using NREL Phase VI horizontal axis wind turbine, Journal of Wind Engineering and Industrial Aerodynamics, Volumes 107–108,).

Also, please provide these three pieces of information about your mesh. Also, please provide the thickness of the boundary layer. This can be calculated using the WOLFDYNAMIC website (http://www.wolfdynamics.com/tools.html?id=110)

Line 293

(from -3 to 2 mm) >>> from -3 mm to 2 mm

Line 417 (Figure 18)

It seems to me that all other figures are generated using TECPLOT, which Figure 18 is generated using EXCEL. I suggest that authors also replot the plots of Figure 18 using the same software used for other plots.

Line 443:

Please mention the Maximum Temperature.

Author Response

ABSTRACT:

1-Since flow control can be used for many applications including wind turbines, airplanes, missiles, etc, I suggest that authors change the word “vehicle” to “aerodynamic object”.

Response 1: The text is corrected

2-The authors should used consistent format for the numbers “Re=1.5x106”

Response 2: A typo in line 141 has been corrected in the text. The Reynolds numbers in the abstract and in the description of the experiment are now equal.

3-Reynolds in a non-dimensional number. Why is the dimension of “ Re “ [1/m]?

Response 3: The unit Reynolds number is defined by the formula Re1=V/nu, where V is the flow velocity and nu is the kinematic viscosity. This number characterizes the properties of the oncoming flow and is often used in scientific papers. For example: Risius, S., Costantini, M., Koch, S. et al. Unit Reynolds number, Mach number and pressure gradient effects on laminar–turbulent transition in two-dimensional boundary layers. Exp Fluids 59, 86 (2018). https://doi.org/10.1007/s00348-018-2538-8.

INTRODUCTION:

Overall, I believe that the INTRODUCTION part should be improved. The authors start with vortex generators which seems reasonable and then provide some information about other methods such as SJAs. In my opinion, it is worth talking about the applications of flow control methods/devices. Hence, I suggest that authors refer to some new references about using flow control methods for different applications. I found the following papers can add value to this research:

Ø Drag Reduction of a Passenger Car Using Flow Control Techniques. Int.J Automot. Technol. 20, 397–410 (2019). https://doi.org/10.1007/s12239-019-0039-2

Ø Design and aerodynamic performance analysis of a finite span double-split S809 configuration for passive flow control in wind turbines and comparison with single-split geometries, Journal of Wind Engineering and Industrial Aerodynamics 214, 104654

Ø CFD analysis of a new active flow control method for wind turbine airfoils using a paddle wheel inside the split of airfoil, J. Comput. Fluids Eng. 26 (2), 69-76

Ø Aerodynamic and Structural Design of a 2022 Formula One Front Wing Assembly. Fluids 2020, 5, 237. https://doi.org/10.3390/fluids5040237

Ø Hypersonic Flow Control of Kinzhal Missile via Off-Axis, Pulsed Energy Deposition Partth A. Laad et al., AIAA Journals, 2021

Ø Magnetohydrodynamic Control of Hypersonic Separation Flows, International Journal of Aerospace Engineering, vol. 2021, Article ID 6652795, 13 pages, 2021.

Response 4: The literature review has been updated to reflect the reviewer's comments.

Also, please consider modifications of the following points:

Line 28:

The references [1-3] are too old. I believe that the authors do not need to go back to the history of flow control. In case they want to mention the importance of flow control, they can simply refer to a recent review paper.

Response 5: References to review papers on this topic have been added. Nevertheless, the mention of pioneer works, from our point of view, makes the review more consistent.

Line 35:

There is little value in listing 5 references [5,6,…,9] just by their number. For example, I suggest authors can only select two papers and provide some highlights and important findings of each of them and explain them briefly. The same comment applies to LINEs  64 , LINE 75 and LINE 91

Response 6: It is true that in the Introduction the results obtained in these works are not discussed in detail. The reason for this is the desire to keep the size of the paper within reasonable limits. Nevertheless, the indication of the papers in the introduction allows us to demonstrate the relevance of this study.

Line 54:

I suggest that authors break the existing paragraph and start a new one from “ A device implementing….”

Response 7: The suggestion is implemented.

SECTION 2

LINE 138:

Please provide the location /owner (laboratory name) of the wind tunnel. Also please list the information of the wind tunnel in a separate Table.

Response 8: A more detailed description of the wind tunnel T-325 is added to the paper. In addition, a reference to the paper with characteristics of this wind tunnel is added as well.

Figure 3:

Is fade. Please replace it with a more clear figure.

Response 9: The figure has been corrected.

 SECTION 3

Line 192:

SST-k-ω is very sensitive to Y+, expansion ratio, and number of cells inside the boundary layer. (Effects of near-wall grid spacing on SST-K-ω model using NREL Phase VI horizontal axis wind turbine, Journal of Wind Engineering and Industrial Aerodynamics, Volumes 107–108,).

Also, please provide these three pieces of information about your mesh. Also, please provide the thickness of the boundary layer. This can be calculated using the WOLFDYNAMIC website (http://www.wolfdynamics.com/tools.html?id=110)

Response 10: The following text is added to the paper:

The thickness of the boundary layer in the region of the actuator was approximately 3 mm. The number of the computational grid cells in the boundary layer was about 55. The value of y+ did not exceed 1. The grid was refined both in the direction of the model wall and in the direction of the wedge. The x+ and z+ values on the wedge surface were about 15÷30. With distance from the wedge, the x size of the cell increased and the value of x+ was about 50. Grid convergence was tested using the wall shear stress as a criterion. The difference in drag between the basic mesh (7.5 million cells) and a more accurate mesh (45 million cells) did not exceed 1%. When comparing the numerical results with the experiment, calculations were carried out using various turbulent models. The numerical results obtained using the κ SST turbulent model showed the best agreement with the experimental data.

Line 293

(from -3 to 2 mm) >>> from -3 mm to 2 mm

Response 11: Corrected

Line 417 (Figure 18)

It seems to me that all other figures are generated using TECPLOT, which Figure 18 is generated using EXCEL. I suggest that authors also replot the plots of Figure 18 using the same software used for other plots.

Response 12: Experimental and numerical data were processed by different people. Experiments - by O.Vishnyakov. Calculations - by P.Polivanov.

The graphs based on the numerical results were obtained using Phyton Matplotlib (IDE Spyder). It would take a lot of time to rebuild them in another software. Therefore, it was decided to change the style of the figure 18 approximating the style of TECPLOT.

Line 443:

Please mention the Maximum Temperature.

Response 13: The text is corrected:

The value of the maximum temperature defect in this section was approximately 130 de-grees for the LE case (Fig. 20c) and 220 degrees for the SP case (Fig. 20d).

Reviewer 2 Report

The authors have studied the boundary layer characteristics in the presence of plasma based vortex generators. The methodology adopted seems good. However, I didn't find any quantification to support their claim. They should look into it.

Author Response

The authors have studied the boundary layer characteristics in the presence of plasma based vortex generators. The methodology adopted seems good. However, I didn't find any quantification to support their claim. They should look into it.

Response 1: Since in this work, flow separation suppression was not investigated, it is not a simple task to evaluate the efficiency of the plasma actuator. It was decided to estimate the efficiency of streanwise vortex generation by the change in the total pressure defect. As a result of calculations, it was obtained that the power spent on heating the flow is approximately equal to the power estimated by formula (1), which is added to the paper.

Reviewer 3 Report

In this drafted manuscript, the authors have employed both numerical simulation and experimental investigation so as to study plasma-based vortex generator in supersonic turbulent boundary layer. There are some concerns that the authors need to address (Major Revision):

-        - No nomenclatures are included (although some are included in the main article). A chart of nomenclature, symbol, subscripts, abbreviations should be prepared in terms of better understanding for readers.

-       - The resolutions of few figures are not readable and too small. They must be reedited.

-        -What is the novelty of paper? What’s the new? What are the differences of this manuscript from Ref. [46, 47, especially 48]? At least, it should be emphasized at the end of the introduction and conclusion part of paper.

-        - Literature research is too weak. It should be expanded with current literature studies including passive flow control techniques.

-        -  Few sentences are not understandable. The writing of the drafted manuscript should be revised by taking into account well English level.

As technically discussing;

-        - Related to the numerical attempt, is there any specific reason of the authors in terms of selecting turbulence model of k-w SST and density-based solver?

-        - The mesh independency is missing. It should be strongly performed in terms of better accuracy.

-        -  How did the author measure the boundary layer (δ)? It should be clearly explained.

-        -  Are the results provided by Fig.4 a calibration study?

-        -  How did the authors put the pitot tube at the wake region? It is missing in the Fig. 3.

In addition to comments mentioned above, the authors are suggested to add actual studies including currently performed passive flow control techniques with their impacts on aerodynamics:

-        -  Experimental flow control investigation over suction surface of turbine blade with local surface passive oscillation

-        -  Experimental investigation on effect of partial flexibility at low aspect ratio airfoil–Part I: Installation on suction surface

-        - Experimental investigation on effect of partial flexibility at low aspect ratio airfoil-Part II: Installation both on suction and pressure surface

-     - Impact of local flexible membrane on power efficiency stability at wind turbine blade

-        - Experimental study of the wind turbine airfoil with the local flexibility at different locations for more energy output

-        -  Mapping of laminar separation bubble and bubble-induced vibrations over a turbine blade at low Reynolds numbers

Overall, the comments and suggestions mentioned above should be taken into consideration carefully for better and scientific paper.

Author Response

There are some concerns that the authors need to address (Major Revision):

- No nomenclatures are included (although some are included in the main article). A chart of nomenclature, symbol, subscripts, abbreviations should be prepared in terms of better understanding for readers.

Response 1: Nomenclature has been added to the paper

- The resolutions of few figures are not readable and too small. They must be reedited.

Response 2: The resolution of the figures has been improved. The font size has been increased in Figures 17-22.

- What is the novelty of paper? What’s the new? What are the differences of this manuscript from Ref. [46, 47, especially 48]? At least, it should be emphasized at the end of the introduction and conclusion part of paper.

Response 3: In the present work, a detailed analysis of the formation of the streamwise vortex is performed. As a result of this analysis, it is shown for the first time that the flow displacement hypothesis does not allow us to fully explain the vortex formation process. It is necessary to carry out additional and more detailed study of this issue. In addition, a detailed comparison of the calculation with the experiment was not performed in the cited works.

- Literature research is too weak. It should be expanded with current literature studies including passive flow control techniques.

Response 4: The literature review has been expanded.

- Few sentences are not understandable. The writing of the drafted manuscript should be revised by taking into account well English level.

Response 5: English has been improved.

As technically discussing;

- Related to the numerical attempt, is there any specific reason of the authors in terms of selecting turbulence model of k-w SST and density-based solver?

- The mesh independency is missing. It should be strongly performed in terms of better accuracy.

Response 6: Density-based solver has better convergence for problems with Mach number M>1. In addition, a text answering the remaining questions has been added to the paper.

- How did the author measure the boundary layer (δ)? It should be clearly explained.

Response 7: The thickness of the boundary layer was estimated from PIV measurements. The point at which the flow velocity was equal to 99% of the oncoming flow velocity was taken as the edge of the boundary layer.

-Are the results provided by Fig.4 a calibration study?

Response 8: Figure 4 shows oscillograms of voltage and current in the discharge obtained in the experiment. The high-frequency discharge was specially used in the study, which made it possible to consider the effect as quasi-stationary. The heat spot generated by the discharge did not have time to leave the area near the wedge before the next moment of ignition of the discharge. This method was used in the work because it is very difficult to generate a static discharge for a high-speed flow. A more detailed description was added to the text.

-How did the authors put the pitot tube at the wake region? It is missing in the Fig. 3.

Response 9: The miniature Pitot tube was located along the flow and mounted on a movable bar in-stead of a prism, which was used in the PIV experiments.

In addition to comments mentioned above, the authors are suggested to add actual studies including currently performed passive flow control techniques with their impacts on aerodynamics:

-Experimental flow control investigation over suction surface of turbine blade with local surface passive oscillation

-Experimental investigation on effect of partial flexibility at low aspect ratio airfoil–Part I: Installation on suction surface

-Experimental investigation on effect of partial flexibility at low aspect ratio airfoil-Part II: Installation both on suction and pressure surface

-Impact of local flexible membrane on power efficiency stability at wind turbine blade

-Experimental study of the wind turbine airfoil with the local flexibility at different locations for more energy output

-Mapping of laminar separation bubble and bubble-induced vibrations over a turbine blade at low Reynolds numbers

Response 10: Some of these works are included in the review.

Round 2

Reviewer 3 Report

Aforementioned comments and suggestions according to the draft version of paper have been understandably clarified by authors. It seems well answered and supported. Thus, that revised form of paper has addressed my concerns well. I suggest the authors provide the accurate and complete authors' information of some references during the proofreading. I recommend that this paper with this form seems to be acceptable for publication in AEROSPACE.